# The Moderating Role of Emotional Regulation on the Relationship between School Results and Personal Characteristics of Pupils with Attention Deficit/Hyperactivity Disorder

**DOI:** 10.3390/children9111637

**Published:** 2022-10-27

**Authors:** Florentina Ionela Linca, Magdalena Budisteanu, Doru Vlad Popovici, Natalia Cucu

**Affiliations:** 1Department of Psychiatry Research Laboratory, ‘Prof. Dr. Alexandru Obregia’ Clinical Hospital of Psychiatry, 041914 Bucharest, Romania; 2Department of Special Psychopedagogy, Faculty of Psychology and Educational Sciences, University of Bucharest, 0506578 Bucharest, Romania; 3Medical Genetics Laboratory, ‘Victor Babes’ National Institute of Pathology, 050096 Bucharest, Romania; 4Department of Preclinical Disciplines, Faculty of Medicine, ‘Titu Maiorescu’ University, 031593 Bucharest, Romania; 5Association for Epigenetics and Metabolomics, 79108 Bucharest, Romania

**Keywords:** ADHD, school results, emotional regulation, problem solving ability, visuospatial integration

## Abstract

This study aimed to explore the possible moderating role of emotional regulation in the relationship between problem-solving ability, visuomotor precision and visuospatial integration on the one hand and school results on the other in pupils with ADHD. A total of 241 pupils with ADHD (study group) and 207 children without ADHD (control group) were included in our research. Specific tests for the evaluation of the problem-solving ability, visuomotor precision, visuospatial integration, and emotional regulation were applied. The results showed that emotional regulation is a significant moderator of the relationship between school results and problem-solving ability, visuomotor precision, visuospatial integration, and working memory. There are statistically significant differences depending on emotional regulation, visuomotor precision, visuospatial integration, problem-solving ability and working memory in terms of school results of students with ADHD compared to children without this diagnosis. These results can be used in the development of intervention programs.

## 1. Introduction

Attention deficit/hyperactivity disorder (ADHD) is characterized by a complex and heterogeneous symptomatology consisting of a persistent pattern of hyperactivity, impulsivity, and inattention [1], and it impacts all aspects of the social life of the person, including family, school and social relationships [2,3,4,5]. Moreover, epidemiological research has found a strong association between ADHD and deficiencies in executive functioning (visuomotor integration problems, poor motor coordination, verbal and non-verbal working memory deficits, and poor problem-solving ability) and emotional regulation [6,7,8,9,10].

In the literature, executive functions are seen as a top-down process and are defined as the abilities that allow the planning and organization of information in working memory and behavioral self-regulation in such a way that the behavioral response is adapted to the dynamic context in which the person is [11]. Meanwhile, emotion regulation is seen as a bottom-up process, and represents the ability of an individual to modify an emotional state in order to achieve adaptive, goal-oriented behaviors [12].

Neuroanatomical and neurofunctional research has shown that although these two systems are different, they interact. In the case of children with ADHD, because we are talking about executive dysfunctions and emotional regulation difficulties, the interactions between emotional and cognitive processes are explained by potential loci of dysfunction [13,14,15,16]. Dysfunctions in the striato-nigral and thalamo-cortico-thalamic networks provide the anatomical basis by which emotional pathways (orbito-medial prefrontal cortex—OMPFC) influence “cold” executive function pathways (dorsolateral prefrontal cortex—DLPFC), which in turn influence the motor function pathways [17,18]. The unidirectional nature of information flowing through the non-reciprocal components of these spiral circuits suggests the robust influence of emotion on cognitive processing, which in turn may significantly influence the motor behavior of children diagnosed with ADHD and related disorders [19,20,21,22,23,24,25].

Over time, studies have been conducted that measured the executive functions and emotional regulation of children with ADHD through questionnaires and neurocognitive tasks [26,27], but there is a need for studies that reflect the unique contribution of both executive functions and emotional regulation to the heterogeneity of ADHD symptoms. Among all of these studies, a holistic approach toward these children is missing on the one hand, and on the other, the objectivity of the survey data to reflect at a behavioral, measurable level the problems of children with ADHD. The personal and clinical characteristics of children with ADHD are highlighted by evidence that would explain the dysfunctions at the brain level in behavioral, measurable terms. Thus, behavioral solutions can be found for a brain dysfunction.

For example, deficits in executive functioning were associated with predominantly inattentive ADHD and with school difficulty, while emotional regulation difficulty was associated with predominantly hyperactive ADHD at preschool age and with predominantly inattentive ADHD at adolescence [26,27,28,29].

Children with ADHD who also have difficulties in emotional regulation are also associated with low social skills, a higher probability that the symptoms of hyperactivity will not remit in adulthood, and a lower quality of life than children without ADHD. Deficits in executive functioning can impair mental processes that support self-regulation and can delay learning and development. Difficulty with emotional regulation was also associated with fewer years of schooling, a lower likelihood of graduating from high school and college, and higher rates of dropping out of school. In addition, emotional regulation could be a protective factor against stress. Children with ADHD who had a high level of emotional self-regulation coped better with stressful situations at school, exams, or in everyday life, as was the case with the COVID-19 pandemic [30,31,32,33].

These contradictory results suggest that there are different associations between cognitive and emotional processes with the heterogeneity of ADHD symptoms throughout a person’s development. Indeed, longitudinal studies have shown that the presentation of ADHD in children varies enormously from the preschool to adolescent years [29]. Instead, it remains unclear to what extent individual differences in executive functioning and emotional regulation contribute to the heterogeneity of ADHD symptoms during childhood.

In other words, because the heterogeneity of ADHD symptoms is very large, researchers have tried to explain it either through the association between cognitive characteristics and certain ADHD symptoms or through associations between emotional characteristics and other ADHD symptoms.

Mohamed et al. [34] explored whether impairments in core processes (processing speed and distractibility) in individuals with ADHD explain impairments in higher order functions, namely executive functions, memory and complex attention. Hierarchical logistic regression analyses to assess the contribution of core processes to impairments in higher-order functions revealed that deficits in core processes explained 41–43% of deficits in executive functions, 27–29% in memory, and 56–74% in complex attention. In this case, the role of emotional regulation skills in performance in timed trials was not taken into account.

To complement the information from the above study, Anker and colleagues [35] showed in their study that the association between the results of neuropsychological tests of verbal working memory and processing speed and the association between the severity of attention deficits and emotional regulation deficits in people with ADHD were significant. A shortcoming of this research was that the problem of executive functioning and emotional regulation in ADHD was not treated in the light of a holistic approach. The variables were grouped sequentially.

Tenenbaum and colleagues [36], as well as the researchers above, demonstrated that difficulty in emotional regulation can be explained by a large number of errors of commission and omission and by parasympathetic dysregulation and reduced sympathetic reactivity.

Morris and colleagues [37] in their research utilized a positive and negative emotion induction and suppression task, as well as indexes of autonomic nervous system reactivity, to examine emotional functioning in youth with ADHD. This study revealed inflexible parasympathetic-based regulation across emotion conditions among youth with ADHD compared to typically developing youth. Future studies should consider the efficacy of adding an emotion regulation skills training component. This study provides the biological basis for the importance of emotional regulation in the functionality of children and young people with ADHD; however, it is only the starting point for the integration of emotional regulation skills in the context of school performance.

All of the mentioned studies approach the emotional regulation and executive functioning of children with ADHD in a restrictive manner. There are studies that show the importance of emotional regulation at the behavioral level by explaining its biological aspects, and there are studies that associate emotional regulation only with certain variables such as working memory. It is necessary to take a holistic approach to all of the functional difficulties of a child with ADHD in such a way that they can integrate and adapt to the school environment.

The present study aimed to explore the possible moderator role of emotional regulation in the relationship between problem-solving ability, visuomotor precision and visuospatial integration on the one hand and school results on the other in pupils with ADHD. To our knowledge, this is the first study which investigated the role of all these factors in the learning process of children with ADHD using objective measures.

Based on the specialized literature, we can state that difficulties in emotional regulation directly determine difficulties in adjusting the emotional state, which can lead to behaviors inconsistent with the achievement of objectives. Consequently, a possible cause of school problems in ADHD students is difficulty using self-regulatory learning behaviors essential for successful academic performance. Thus, we can justify the moderating role of emotional regulation in the relationship between school results and the personal characteristics of students with ADHD [30,38,39,40].

Interestingly, inattentive and hyperactive/impulsive symptoms both uniquely predicted emotion regulation, even when controlling for executive functioning, and 18–30% of working memory’s relation with emotion regulation was conveyed via shared associations with these ADHD symptoms. This pattern suggests that working memory is related to emotion regulation at least in part because underdeveloped working memory contributes to the development and severity of ADHD symptoms, which, in turn, predicts emotion dysregulation [41,42].

Instead, neuroimaging studies suggest that increased attention-related neural activity in emotional contexts reflects the allocation of cognitive resources for regulatory control. Such results indicate that children’s difficulties in regulating behaviors in emotional contexts may be the result of competition between well-developed emotional processes and poorly developed cognitive control systems [43].

## 2. Materials and Methods

### 2.1. Patient Data

The study was performed as a doctoral study of the Faculty of Psychology of the University of Bucharest during the year 2016, and included two groups of children: (1) children diagnosed by a psychiatrist with ADHD from a regular school (study group) and (2) children without ADHD (control group). The inclusion criteria for the children from the first group were: diagnosis of ADHD established by a child psychiatrist based on DSM 5 criteria [1]; age between 6 and 12 years; IQ 75–100; and absence of other neurodevelopmental disorders. The control group included children without any neuropsychiatric conditions, matched for school results, IQ, age and gender with study group.

The school results of the pupils did not reveal any school difficulties. An objective measurement of learning difficulties was represented by pupils’ grades. Meanwhile, after the examination, the psychiatrist established that none of the children included in our groups had learning difficulties.

We chose an intelligence quotient between 75 and 100 because studies have illustrated the fact that the low or very high level of intelligence of children with ADHD determines the emergence of social integration problems, difficulties in coordinating movements and difficulties in verbal working memory. In addition, the hyperactivity of very intelligent children is explained by the high speed of information processing compared to children of the same chronological age, but with typical development [44].

In addition, children and adults with high intelligence quotients (IQs) had lower levels of ADHD symptom severity and lower odds of having executive functioning problems that are often found in people with ADHD [45,46,47].

### 2.2. Methods and Materials

In order to measure the level of attention deficit and hyperactivity, the Romanian version of the Teacher Rating Scale/TRS-P from BASC-2 (Behaviour Assessment System for Children-2) developed and standardized by [48] for 6–11-year-old children was used. Teachers assessed the behavior of children with attention deficit and hyperactivity disorder for a period of 6 months on a 4-point scale: never (0), sometimes (1), often (2) and always (3). Item raw and subscale t scores from the 2 primary areas were utilized (attention problems and hyperactivity). Raw scores represented the total points for all items. These scores were reported as t scores with a mean of 50 and a standard deviation of 10.

For the assessment of the neuropsychological functioning (attention/executive functioning, visuospatial, sensoriomotor, and memory), the Neuropsychological Development Assessment (NEPSY) was used [49]. NEPSY is a tool that evaluates key functions of a child for higher performances both in school and out of school and applies to all children aged 3 to 12 years. In this study, we used the following evaluation methods: tower, which measured the problem-solving ability; design copying, which measured the visuomotor integration; visuomotor precision, which measured the precision and accuracy of fine digital movements and memory for faces, which measured the level of development of working memory. Standard scores between 1 and 4 were well below the expected level, standard scores between 5 and 6 were below the expected level, a standard score of 7 was at the limit, scores between 8 and 9 were at the expected level, and standard scores greater than 10 were above the expected level.

We chose to test these components of executive function because they constitute the basis of the organization and planning of the learning process. If executive dysfunctions occur, attention difficulties and difficulties in planning and organizing the problem-solving process may occur, and these represent the basis of issues with reading, writing and simple mathematical calculation. Thus, learning difficulties may appear. The child, not paying attention to what they read, will lose the meaning of the words read, and thus will not be able to understand the task they have to solve. Fundamentally, the learning difficulties are based on instrumental deficits (attention deficits and working memory deficits) [6,7,8,9,10].

Following the differential diagnosis of a learning disorder and ADHD, we notice that some difficulties appear only in a certain context and, therefore, the symptoms are better explained by ADHD than by a learning disorder. Difficulty paying attention and planning activities regardless of nature is specific to ADHD. Problems can appear in the chain, but it is basically ADHD that causes them.

In order to identify difficulties regarding emotional regulation (difficulties in recognizing facial expressions, identifying emotions, identifying emotional responses), an imaging system was developed.

This imaging system was validated in a group of 230 children with neurotypical development before its application to the samples in this study. The fidelity expressed by the Cronbach’s alpha index was 0.87, which allowed its use in this study. The test–retest reliability indicated a value of 0.83 at an interval of 6 months.

The test contained 60 items grouped into 3 scales—recognition of facial expressions, identification of emotions and identification of emotional response. Content validity was ensured by the selection of images/items based on studies in the field (the relationship between brain and behavior, the identification and understanding of emotions and the social contexts in which they appear) by two experts in child psychological assessment and child development [30,34,35,36,37]. Content adjustment was based on the same principle as selection. The subsequent improvements of the test were based on the clinical and scientific experience gained by the authors with a series of children. This instrument was revised very carefully 3 times in terms of content, and after the pilot test, it was included in this study and the psychometric characteristics were established.

Construct validity was established by checking the correlations between the items of the subscales and subscales that measure similar contents. In our pilot group, there were medium correlations (0.4–0.6) between subscales and high correlations between the items of each subscale with the respective subscale (0.7–0.8). The pattern of correlations between each subscale and its items provided clear evidence that the structure of the entire test was solid.

Recognition of facial expressions—The pupil looked at a picture of a facial expression and then identified the same expression given a choice of four pictures.

Identification of emotions—The pupil identified an emotion from a choice of four pictures.

Identification of emotional response—The pupil matched a picture of an emotion to a picture that elicited that emotion from a choice of four pictures. Each subtest contained 20 items.

A global score was calculated. A score of 1–16 points was defined as a low score, a score of 17–32 points was an average score, and a score of 33–48 points was a high score. These scores were established based on calculated percentiles. It was important to establish performance levels based on percentiles, because in this way we could observe where the child is after a test.

Emotion regulation is a component of the broader concept of self-regulation. Predescu et al. [50] claimed that self-regulation refers primarily to the control and regulation of one’s emotions and overlaps substantially with inhibitory control, an important component of executive functions.

The test was based on the model of emotional regulation as a process developed by Gross in 2002. This model identifies five emotion regulation strategies that occur during different time points in the emotion experience: situation selection, situation modification, attentional deployment, cognitive change, and response modulation. Gross (1998) further divided these strategies into antecedent-focused and response-focused strategies. Antecedent-focused regulation occurs before the emotion is fully experienced or during the emotion experience, whereas response-focused regulation occurs after the emotion has completely developed. So, with response-focused regulation, people have already “responded” to the eliciting event and thus have experienced all the emotion component changes. Within response-focused regulation, people can regulate their emotions by trying to change any of the emotion components. They might change their facial expressions and vocal tone, suppress their thoughts, increase or decrease their physiological arousal, and even change their subjective feelings [51]. Therefore, recognizing and identifying emotions in our own person and in the people around us helps us to regulate our behavior before and after the emotion has appeared.

A similar test was carried out by Wiing in 2008 [52], Social Emotional Evaluation, which, in addition to the scales we mentioned, also tested the understanding of different conflicting social contexts (understanding social gaffes—the student looks at a picture of a social situation and listens to an accompanying audio clip, then he/she identifies whether everyone in the situation behaved appropriately; understanding conflicting messages—the student looks at a picture and listens to the accompanying audio clip. He/she determines whether the situation contains a conflicting message, and if so, identifies the true meaning of the message. The conflicting messages include humor, sarcasm, and lies). Test–retest reliability was 0.88.

All these tests involve performing tasks, an experimental process that establishes a level of development of the tested area or the severity of symptoms.

The consent of the Ethics Commission of the University of Bucharest was obtained for this study on the one hand, but also the consent of the parents of the children included in this study and of the principals of the schools where the children studied on the other. An informed consent form was completed and signed by each parent of the participating children. Participation was voluntary, and children were selected from schools in Bucharest where the researchers had the consent of the principal and the children’s parents. All applicable international, national and/or institutional guidelines for the care of the children were followed.

### 2.3. Statistical Analysis

For statistical analysis, a moderation test, a Spearman test and a U Test were performed. The moderation and Spearman test were applied to verify the relationships between the research variables, and the U Test was applied to verify the differences between the group of children with ADHD and the control group. In addition, the moderator is the variable that can modify, intensify or decrease the relationship between two other variables. We used JASP 0.16.2.0 software.

## 3. Results

### 3.1. Analysis of Participant Demographic and Clinical Data

#### 3.1.1. Differences between the Control Group and the Research Group in Terms of Personal Characteristics and School Results

The study included 241 children with ADHD (154 boys and 87 girls) aged between 6 and 10 years (M = 97.5 months/8 years and 2 months, SD = 15) (study group) and 207 children without ADHD (66 girls and 141 boys) aged between 6 years and 10 years 4 months (M = 96 months/8 years, SD = 14.7) (control group). None of the children with ADHD received medication and/or psychotherapy during the study. Overall, 74 (30.8%) children with ADHD had very poor school results, 128 (53.1) children with ADHD had poor school results and 39 (16.1%) children had average school results. In the control group, 12 (5.8%) children had very poor school results, 157 (75.8%) children had poor school results, and 38 (18.4%) children had average school results (Table 1). Very poor school results correspond to insufficient grades, poor school results correspond to sufficient grades, and average school results correspond to good grades.

These children do not have school difficulties or a low level of intelligence. School results can be explained by a multitude of factors. We did not consider pathology in all cases with poor school results. We chose the control group with the same characteristics as the study group.

Regarding the problem-solving ability in children with ADHD, the tower test revealed scores well below the expected level in 47 (19.5%) cases, scores below the expected level in 62 (25.7%) children, scores at the limit in 25 (10.4%) cases, scores at the expected level in 67 (27.8%) children and scores above the expected level in 40 (17.6%) children. There was a statistically significant positive relationship between problem-solving ability and school results (r = 0.395, *p* < 0.01) in the study group. For the control group, the same test revealed scores well below the expected level in 8 (4%) children, scores below the expected level in 20 (9.6%) cases, scores at the limit in 41 (19.8%) children, scores at the expected level in 73 (35.2%) children and scores above the expected level in 65 (31.4%) children. In this case, problem-solving ability correlated with school results (r = 0.194, *p* < 0.01). There were statistically significant differences between children in the study group and children in the control group depending on problem-solving ability in terms of school results (X^2^(2) = 666.450, *p* < 0.001) (Table 1 and Table 2).

In the study group, the following results were obtained in the visuomotor precision test: 76 (31.5%) children with ADHD had scores well below the expected level, 75 (31%) children had scores below the expected level, 32 (13.2%) children had scores at the limit, 34 (15%) children had scores at the expected level and 24 (9.3%) children had scores above the expected level. Visuomotor precision correlated with school results (r = 0.383, *p* < 0.01) in the study group. Meanwhile, the control group in the same sample had the following results: 19 (9.2%) children had scores well below the expected level, 46 (22.2%) children had scores below the expected level, 46 (22.2%) children had scores at the limit, 41 (19.8%) children had scores at the expected level and 55 (26.6%) children had scores above the expected level in terms of visuomotor precision. A statistically significant positive relationship between visuomotor precision and school results (r = 0.517, *p* < 0.01) was found for the control group. There were statistically significant differences between children in the study group and children in the control group depending on the visuomotor precision in terms of school results (X^2^(2) = 1455.220, *p* < 0.001) (Table 1 and Table 2). The children in the study group had poorer results than the children in the control group in terms of school results and visuomotor precision.

The study of visuospatial integration in children with ADHD showed that 58 (24%) children had scores well below the expected level, 52 (21.5%) children had scores below the expected level, 24 (10%) children had scores at the limit, 44 (18.2%) children had scores at the expected level and 63 (27.3%) children had scores above the expected level. In the study group, there was a statistically significant positive relationship between visuospatial integration and school results (r = 0.345, *p* < 0.01). In the control group, 24 (11.6%) children had scores well below the expected level, 30 (14.6%) children had scores below the expected level, 19 (9.1%) children had scores at the limit, 45 (21.7%) children had scores at the expected level and 89 (43%) children had scores above the expected level. The school results positively correlated with visuomotor integration (r = 0.415, *p* < 0.01) in the control group. There were statistically significant differences between the children in the study group and the children in the control group based on the visuomotor integration in terms of school results (X^2^(2) = 3539.133, *p* < 0.001) (Table 1 and Table 2).

Regarding the working memory in children with ADHD, 96 (39.8%) children had scores well below the expected level, 96 (39.8%) children had scores below the expected level, 26 (10.8%) children had scores at the limit, 20 (8.2%) children had scores at the expected level and 3 (1.4%) children had scores above the expected level. A statistically significant correlation between working memory and school results was observed (r = 0.355, *p* < 0.01) in the study group. In the control group, 6 (2.5%) children had scores well below the expected level, 36 (17.9%) children had scores below the expected level, 33 (15.9%) children had scores at the limit, 88 (42.4%) children had scores at the expected level and 44 (22.3%) children had scores above the expected level. The working memory correlated positively with school results (r = 0.406, *p* < 0.01) in the control group. There were statistically significant differences between the children in the study group and the children in the control group based on working memory in terms of school results (X^2^(2) = 1509.128, *p* < 0.001) (Table 1 and Table 2).

The scores obtained by children with ADHD in the emotional regulation test were low in 163 (67.2%) children and average and high in 39 (16.4%) children. Emotional regulation correlated positively with school results (r = 0.156, *p* < 0.01) in the case of children with ADHD. In the control group, 36 (17.3%) children had low scores, 63 (30.4%) children had average scores and 108 (52.3%) children had high scores. There was a statistically significant relationship between emotional regulation and school results (r = 0.309, *p* < 0.01) in the control group. The children from the control group regulated their emotions better than the children with ADHD (X^2^(2) = 1300.936, *p* < 0.001) (Table 1 and Table 2).

#### 3.1.2. The Moderating Role of Emotional Regulation on the Relationship between Personal Characteristics and School Results of Children with/without ADHD

Regarding the moderating role of emotional regulation in the relationship between school results and visuomotor precision of children with/without ADHD, the data were statistically significant (14.1 (4.56), *p* < 0.01 for the study group and −1.38 (8.19), *p* < 0.05 for the control group) (Table 3 and Table 4). If children with ADHD have problems with emotional regulation, then they will have poor school results and poor performance in visuomotor precision tests. In the case of children in the control group, good emotional regulation skills can be associated with good results at school and good results in the visuomotor precision test. We can conclude that the difficulties in emotional regulation in relation to the difficulties in visuomotor precision and poor results at school are characteristic for children with ADHD.

The relationship between school results and the visuospatial integration of pupils with/without ADHD was moderated by emotional regulation, and the results were statistically significant (15.4 (3.79), *p* < 0.01 for the study group and −1.91 (9.43), *p* < 0.01 for the control group) (Table 3 and Table 4). If children with ADHD have problems with emotional regulation, then they will have poor school results and poor performance in visuospatial integration tests. We can conclude that the difficulties in emotional regulation in relation to the difficulties in visuospatial integration and poor results at school are characteristic for children with ADHD.

The moderating role of emotional regulation in the relationship between the school results and working memory of children with/without ADHD was statistically significant (10.9 (6.08), *p* < 0.05 for the study group and 3.53 (0.001), *p* < 0.01 for the control group). If children with ADHD have problems with emotional regulation, then they will have poor school results and poor performance on working memory tests. We can conclude that the difficulties in emotional regulation in relation to the difficulties in working memory and poor results at school are a characteristic of children with ADHD.

Although in all three of the above situations emotional regulation proved to be a moderator for all relationships at all levels, children from the study group had significantly lower scores than children in the control group (Table 2), which leads us to conclude that the difficulties in emotional regulation affect both school performance and test performance.

The moderator can maximize or minimize the association between two variables. Emotional regulation has an impact on the level of all other variables.

Emotional regulation has a statistically significant moderating role in the relationship between school results and problem-solving ability of children with ADHD (9.07 (9.93), *p* < 0.01 for the study group and 4.46 (0.001), *p* < 0.01 for the control group). If children from the study group and from the control group have problems with emotional regulation, then they will have poor school results and poor performance in problem-solving ability tests. Table 3 and Table 4 show that emotional regulation has effects on the school results of children with ADHD when variables have low, average and high values, and in the case of children from the control group, emotional regulation has effects on school results only when both variables have high and average values.

## 4. Discussion

This study demonstrates that emotional regulation is a significant moderator in relation with school results on the one hand and problem-solving ability, visuomotor precision, visuospatial integration, and working memory on the other. Additionally, there was a significant difference in terms of the school performance of pupils with ADHD compared to pupils without this condition depending on problems with emotional regulation, visuomotor precision, visuospatial integration and working memory. If a pupil with ADHD has big problems in emotional regulation, then they will have poorer results at school, but also in the tests that measure the accuracy and precision of fine digital movements, sensory integration, attention and executive functions.

These results are consistent with the results of previous studies. For instance, Brossard-Racine [24] showed that emotions influence the ability to pay attention to details, which influences the quality of motor behavior [53]. The studies conducted by Ghanizadeh and Mous and colleagues [54,55] showed that fine motor skills were predicted by the severity of symptoms in children with ADHD. Motor responses require attention on a target and attention during the response. If the target is not properly focused on, it will affect the subsequent motor planning and consequent performance. Additionally, when the target is not noticed in time, it can reduce the time remaining for motor preparation and accordingly affect the performance [56].

All this is also explained by the phenomena that occur in the brain of children with this condition. There is an anatomical basis through which emotional pathways (orbito-medial prefrontal cortex—OMPFC) influence “cold” executive function pathways (dorsolateral prefrontal cortex—DLPFC), which in turn influence the motor function pathways [17,18]. The unidirectional nature of information flowing through the non-reciprocal components of these spiral circuits suggests the robust influence of emotion on cognitive processing, which in turn may significantly influence the motor behavior of children diagnosed with ADHD and related disorders [19,20,21,22,23,24,25].

Racine et al. [56] showed that a child with ADHD has many problems, such as difficulties in visuomotor precision, visuospatial integration, and working memory; all these influence school activity and results. Poor school results can lead to a very high level of frustration in a child with ADHD, which can also lead to dysfunctional behavior. In our study, all these problems occurred more often in children with ADHD than in the control group. Following the differential diagnosis of learning disorders and ADHD, we notice that some difficulties appear only in a certain context and, therefore, the symptoms are better explained by ADHD than by a learning disorder. Difficulty paying attention and planning activities regardless of nature is specific to ADHD. Problems can appear in the chain, but it is basically ADHD that causes them.

Regarding the moderating role played by the emotional regulation in the association between school results and the characteristics of children with ADHD (problem-solving ability, visuospatial integration, visuomotor precision and working memory), there are very few studies that have been published so far. Most studies took into account age, gender, type of ADHD, comorbidities, severity and comorbidity of problem behavior (ADHD symptoms, conduct and internalizing problems), social functioning, and classroom variables (teaching experience, class size) in the role of moderator [24,57,58]. Veenman and colleagues [59] assessed which moderators influenced the effectiveness of a low-intensity behavioral teacher program for children with symptoms of attention-deficit/hyperactivity disorder (ADHD). In this study, primary school children (*n* = 114) with ADHD symptoms in the classroom were randomly assigned to the intervention program (*n* = 58) or control group (*n* = 56). Moderators included demographic characteristics (gender, age, parental educational level), severity and comorbidity of problem behavior (ADHD symptoms, conduct and internalizing problems), social functioning, and classroom variables (teaching experience, class size).

## 5. Conclusions

In conclusion, as seen in previous studies, difficulties in inhibiting or postponing a certain emotional response, difficulties in developing an action plan, difficulties in sequencing actions and difficulties in forming a mental representation of tasks through working memory influence the accuracy and precision of the movements, and these are reflected in the school results, especially in handwriting tasks or complex tasks that require the completion of several steps in order [58,60].

Through the present study, we have illustrated the specific role of emotional regulation in the relationship between the personal characteristics of pupils with ADHD and their school results. In addition, up to this moment, to our knowledge, there is no study that has tested emotional regulation using a practical test in the complex context of children with ADHD. At the same time, holistic approaches toward children with ADHD in the school context have not appeared in the research so far. Unlike other studies, our study took into account all the emotions that children experience in the school environment, and the executive functions were tested with neuropsychological tests. We believe that emotions support the behavior of organizing and planning school activities, and all these are supported by the statistical significance of the moderation model in this study. On the other hand, the biological basis of emotional regulation supports it in its role as moderator of the relationship between the personal characteristics and school results of pupils with ADHD.

### 5.1. Theoretical and Practical Significance

Our study provides a holistic perspective on the impact of all problems associated with ADHD on school performance of these children. We demonstrated that difficulties in visuomotor precision, visuospatial integration, and working memory, in association with difficulties in emotional regulation, negatively influence the school results of children with ADHD. Accordingly, all these factors must be taken into consideration simultaneously in the management plan of these children.

### 5.2. Limitations and Directions for Future Studies

Some limitations of this study must be noted. The major one is its cross-sectional design. Although the direction of the relationships is theoretically founded and supported by previous studies (e.g., [12]), future research should longitudinally investigate all presumed relationships, especially with respect to the moderation hypothesis. Secondly, this study was also based on the school results of the pupils, which can sometimes be established on subjective criteria, depending on the perspective of the teaching staff; the usual variance of the activity product analysis method may partially explain some of the results. It could be advantageous to obtain multiple measures of some constructs (e.g., engagement from the teachers’ perspective). 

Therefore, good emotional adjustment can maximize the relationship between school results and personal characteristics of children with ADHD. The regulation of the emotional state can determine the planning and organization of behavior in achieving school objectives. Thus, emotional regulation can be seen as a protective factor and as a basic resource in the motivation for school performance. In this way, we answer many questions related to the specific role of emotional regulation in the learning process. Teachers, knowing all these things, can work on improving the level of motivation of children for school tasks and on improving the level of self-confidence of children with ADHD, who are often affected by social stigma.

## Figures and Tables

**Table 1 children-09-01637-t001:** Characteristics of the participants.

		Study Group		Control Group	
		Counts	% of Total	Counts	% of Total
Gender	Boys	154		141	
	Girls	87		66	
Age		M = 97.5 months/8 years and 2 months, SD = 15		M = 96 months/8 years, SD = 14.7	
School results	very poor	74	30.8%	12	5.8%
	poor	128	53.1%	157	75.8%
	average	39	16.1%	38	18.4%
problem-solving ability	well below the expected level	47	19.5%	8	4%
	below the expected level	62	25.7%	20	9.6%
	at the limit	25	10.4%	41	19.8%
	at expected level	67	27.8%	73	35.2%
	at above expected level	40	17.6%	65	31.4%
Visuomotor precision	well below the expected level	76	31.5%	19	9.2%
	below the expected level	75	31%	46	22.2%
	at the limit	32	13.2%	46	22.2%
	at expected level	34	15%	41	19.8%
	at above expected level	24	9.3%	55	26.6%
Visuomotor integration	well below the expected level	58	24%	24	11.6%
	below the expected level	52	21.5%	30	14.6%
	at the limit	24	10%	19	9.1%
	at expected level	44	18.2%	45	21.7%
	at above expected level	63	27.3%	89	43%
Working memory	well below the expected level	96	39.8%	6	2.5%
	below the expected level	96	39.8%	36	17.9%
	at the limit	26	10.8%	33	15.9%
	at expected level	20	8.2%	88	42.4%
	at above expected level	3	1.4%	44	22.3%
Emotional regulation	low scores	163	67.2%	36	17.3%
	average scores	39	16.4%	63	30.4%
	high scores	39	16.4%	108	52.3%

**Table 2 children-09-01637-t002:** Differences depending on emotional regulation, visuomotor precision, visuospatial integration, problem-solving ability and working memory in terms of school results of pupils with ADHD compared to children without this diagnosis.

	Value	df	*p*
Χ^2^ _emotional regulation_	1300.936	2	<0.001
Χ^2^ _visuomotor precision_	1455.220	2	<0.001
Χ^2^ _visuomotor integration_	3539.133	2	<0.001
Χ^2^ _problem solving ability_	666.450	2	<0.001
Χ^2^ _working memory_	1509.128	2	<0.001

**Table 3 children-09-01637-t003:** The effect of the predictors (problem-solving ability, visuomotor precision, visuomotor integration, working memory) on the dependent variable (school results) at different levels of the moderator (emotional regulation) for the study group.

	Estimate	SE	Z	*p*
Problem-solving ability	0.22526	0.01307	17.23	<0.001
Emotional regulation	−0.03550	0.00236	−15.03	<0.001
Problem-solving ability × emotional regulation	0.00901	9.93114	9.07	<0.001
Average	0.225	0.0151	14.88	<0.001
Low (−1 SD)	0.107	0.0204	5.22	<0.001
High (+1 SD)	0.344	0.0211	16.34	<0.001
Visuomotor precision	0.12963	0.00529	24.5	< 0.001
Emotional regulation	−0.04944	0.00219	−22.6	<0.001
Visuomotor precision × emotional regulation	0.00642	4.56124	14.1	<0.001
Average	0.1296	0.00760	17.06	<0.001
Low (−1 SD)	0.0450	0.00971	4.63	<0.001
High (+1 SD)	0.2143	0.01110	19.31	<0.001
Visuospatial integration	0.10498	0.00440	23.9	<0.001
Emotional regulation	−0.05833	0.00245	−23.8	<0.001
Visuospatial integration × emotional regulation	0.00584	3.79114	15.4	<0.001
Average	0.1050	0.00663	15.84	<0.001
Low (−1 SD)	0.0280	0.00818	3.42	<0.001
High (+1 SD)	0.1820	0.00976	18.64	<0.001
Working memory	0.12867	0.00775	16.6	<0.001
Emotional regulation	−0.03570	0.00245	−14.6	<0.001
Working memory × emotional regulation	0.00661	6.08444	10.9	<0.001
Average	0.1287	0.00957	13.44	<0.001
Low (−1 SD)	0.0416	0.01191	3.49	<0.001
High (+1 SD)	0.2158	0.01419	15.21	<0.001

SE: Standard error.

**Table 4 children-09-01637-t004:** The effect of the predictors (problem-solving ability, visuomotor precision, visuomotor integration, working memory) on the dependent variable (school results) at different levels of the moderator (emotional regulation) for the control group.

	Estimate	SE	Z	*p*
Problem-solving ability	0.04776	0.01499	3.19	0.001
Emotional regulation	−0.02552	0.00451	−5.66	<0.001
Problem-solving ability × emotional regulation	0.00834	0.00187	4.46	<0.001
Average	0.0478	0.0156	3.057	0.002
Low (−1 SD)	−0.0158	0.0236	−0.669	0.503
High (+1 SD)	0.1113	0.0189	5.890	<0.001
Visuomotor precision	0.05836	0.00662	8.82	<0.001
Emotional regulation	−0.02195	0.00416	−5.28	<0.001
Visuomotor precision × emotional regulation	−0.00114	8.19324	−1.39	0.044
Average	0.0584	0.00665	8.78	<0.001
Low (−1 SD)	0.0670	0.00949	7.06	<0.001
High (+1 SD)	0.0497	0.00874	5.68	<0.001
Visuospatial integration	0.04135	0.00670	6.17	<0.001
Emotional regulation	−0.02145	0.00471	−4.56	<0.001
Visuospatial integration × emotional regulation	−0.00180	9.43494	−1.91	0.046
Average	0.0414	0.00677	6.11	<0.001
Low (−1 SD)	0.0551	0.01043	5.28	<0.001
High (+1 SD)	0.0276	0.00932	2.96	0.003
Working memory	0.10377	0.01294	8.02	<0.001
Emotional regulation	−0.03703	0.00444	−8.34	<0.001
Working memory × emotional regulation	0.00586	0.00166	3.53	<0.001
Average	0.0414	0.00677	6.11	<0.001
Low (−1 SD)	0.0551	0.01043	5.28	<0.001
High (+1 SD)	0.0276	0.00932	2.96	0.003

## Data Availability

The datasets used and/or analyzed during the current study are available from the corresponding author on reasonable request.

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
