# Peer review of "The Moderating Role of Emotional Regulation on the Relationship between School Results and Personal Characteristics of Pupils with Attention Deficit/Hyperactivity Disorder"

_children, 2022, doi:10.3390/children9111637_

Round 1
Reviewer 1 Report
Correct in abstract: "with ADHD compared to children without this diagnosisThese results can be used in the development of intervention programs.
Introduction: covers aspects of ADHD, although no mention is made of the relationship between learning difficulties and school performance or school results.
Method: Baseline executive functions are assessed, but academic outcomes are not alluded to.
Conclusions: They are scarce, it would be necessary to expand and connect more with what has been done.
Author Response
Correct in abstract: "with ADHD compared to children without this diagnosisThese results can be used in the development of intervention programs.
I corrected the text
Introduction: covers aspects of ADHD, although no mention is made of the relationship between learning difficulties and school performance or school results.
Children with ADHD who also have difficulties in emotional regulation are also associated with low social skills, a higher probability that the symptoms of hyperactivity will not remit in adulthood, a lower quality of life than children without ADHD. Deficits in executive functioning can impair mental processes that support self-regulation and can delay learning and development. Difficulty with emotional regulation was also associated with fewer years of schooling, lower likelihood of graduating from high school and college, and higher rates of dropping out of school. In addition, emotional regulation could be a protective factor against stress. Children with ADHD who had a high level of emotional self-regulation coped better with stressful situations at school, exams, or in everyday life, as was the case with the COVID-19 pandemic [30–33].
Based on specialized literature, we can state that difficulties in emotional regulation directly determine difficulties in adjusting the emotional state, which can lead to behaviors inconsistent with the achievement of objectives. Consequently, a possible cause of school problems in ADHD students is difficulty using self-regulatory learning behaviors essential for successful academic performance. Thus, we can justify the moderator role of emotional regulation in the relationship between school results and personal characteristics of students with ADHD [38-41].
Interestingly, inattentive and hyperactive/impulsive symptoms both uniquely predicted emotion regulation, even when controlling for executive functioning, and 18%-30% of working memory’s relation with emotion regulation was conveyed via shared associations with these ADHD symptoms. This pattern suggests that working memory is related to emotion regulation at least in part because underdeveloped working memory contributes to the development and severity of ADHD symptoms, which, in turn, predicts emotion dysregulation[42, 43].
Instead, neuroimaging studies suggest that increased attentionrelated neural activity in emotional contexts reflects allocation of cognitive resources for regulatory control. Such results indicate that children’s difficulties in regulating behaviors in emotional contexts may be the result of competition between welldeveloped emotional processes and poorly developed cognitive control systems[44, 45].
Method: Baseline executive functions are assessed, but academic outcomes are not alluded to.
We chose to test these component51of the executive functions, because they constitute the basis of the organization and planning of the learning process. If executive dysfunctions occur, we are talking about attention difficulties, difficulties in planning and organizing the problem-solving process, and all these represent the basis of reading, writing and simple mathematical calculation. Thus, learning difficulties may appear. The child, not paying attention to what he reads, will lose the meaning of the words read and thus will not be able to understand the task he has to solve. Fundamentally, the learning difficulties are based on instrumental deficits (attention deficits, working memory deficits) [6–10].
Conclusions: They are scarce, it would be necessary to expand and connect more with what has been done.
In conclusion, as it appears from previous studies, difficulties in inhibiting or postponing a certain emotional response, difficulties in developing an action plan, difficulties in sequencing actions and difficulties in forming a mental representation of tasks through working memory influence the accuracy and precision of the movements and these are reflected in the school results, especially in handwriting tasks or complex tasks that require the completion of several steps in order[58, 59].
Through the present study, we have illustrated the specific role of emotional regulation in the relationship between the personal characteristics of pupils with ADHD and their school results. In addition, up to this moment, to our knowledge, there is no study that has tested emotional regulation using a practical test in the complex context of the child with ADHD. At the same time, the holistic approach of the child with ADHD in the school context does not appear in the research so far. Unlike other studies, our study took into account all the emotions that the child has in the school environment, and the executive functions were tested with neuropsychological tests. We believe that emotions support the behavior of organizing and planning school activities, and all these are supported by the statistical significance of the moderation model in this study. On the other hand, the biological basis of emotional regulation supports it in its role as moderator of the relationship between personal characteristics and school results of pupils with ADHD.
Theoretical and Practical Significance
Our study brings a holistic perspective on the impact of all problems associated with ADHD on school performance of these children. We demonstrated that difficulties in visuomotor precision, visuospatial integration, and working memory, in association with difficulties in emotional regulation negatively influence the school results of children with ADHD. Accordingly, all these factors must be taken into consideration simultaneously in the management plan of these children.
Limitations and Directions for Future Studies
Some limitations of this study must be noted. The major one is its cross-sectional design. Although the direction of the relationships is theoretically founded and supported by previous studies (e.g., (12)), future research should longitudinally investigate all presumed relationships especially with respect to the moderation hypotheses. Secondly, this study was also based on the school results of the pupils, which can sometimes be established on subjective criteria, depending on the perspective of the teaching staff; the usual variance of the activity product analysis method may partially explain some of the results.It could be an advantage to obtain multiple measures of some constructs (e.g., engagement from the teachers’ perspective).
Therefore, a good emotional adjustment can maximize the relationship between school results and personal characteristics of children with ADHD. The regulation of the emotional state can determine the planning and organization of behavior in achieving school objectives. Thus, emotional regulation can be seen as a protective factor, as a basic resource in the motivation for school performance. In this way, we answer many questions related to the specific role of emotional regulation in the learning process. Teachers, knowing all these things, can work on improving the level of motivation of children for school tasks and on improving the level of self-confidence of children with ADHD, who are often affected by social stigma.
Reviewer 2 Report
Thank you for giving us an opportunity to review your research.
Author Response
Thank you very much!
Reviewer 3 Report
1. Potential challenges of the rigidness and consistency of this draft
Comparing with the research goal (Line 17 - Line 19), the title of this draft is
relatively vague. To spotlight the core of this draft, it is strongly suggested that the
term “emotional regulation” should be added into the paper title.
The review (Line 46 – Line 65) does not clear disclosure the research gaps or the socalled “contradictory results” owing to lacking of direct literature support. As well,
most of the cited studies (i.e., Line 26 – Line 29) are not those studies published
recently. It then raises the challenge that whether the authors have reviewed the
studies in this filed thoroughly, and that whether the recent literature still regards this
proposed viewpoint (rather alternative viewpoints) in drawing the casual links
between school results and personal characteristics of pupils with ADHD.
As well, it is also hard to see why emotional regulation plays an important role and
why it is suggested to be put as the moderator, owing to lacking of direct theoretical
support.
2. Problems about the reliability and validity of the proposed measures
According to the draft, the image system is developed by the authors (Line 117 -
Line124). Although the authors addressed that the measures were validated in a
group of 230 children before applying to this study, the reliability and validity (i.e.,
construct reliability, content reliability) of the proposed scales were not reported,
however.
In addition, the appropriateness of the proposed approach of cutting points and
classification results (i.e., segmenting into three levels as stated in Line 119-120) is
also challenged.
3. The role / importance of “emotional regulation” is vague, thus calling for more
explanations and supports
Based upon current findings, it is hard to tell whether / why there exist similar
patterns / impacts of emotional regulation on the relationship between different
school results and personal characteristics of pupils with ADHD.
In addition, it is also challenging to see whether / why emotional regulation does
serve the role of mediator (rather the role of amplifier) for the causality between
school results and personal characteristics of pupils with ADHD.
4. Key contributions have to be drawn more rigidly
According to Line 330-Line335, it is hard to see significant findings reacting to
current studies.
In addition, the authors do not clearly state how the proposed findings of this draft
outperform the findings of current studies, and how they react to the proposed
research gap and claims of current literature.
Moreover, no clear theoretical or practical implications are proposed in the end of
the draft
5. A minor issue for collection
It is strange to see lots of words embedded with “-“ without any meaningful reason
(i.e., Line 36 “re-search”, Line 37 “func-tioning”, Line 39 “regula-tion”)

Author Response
- Potential challenges of the rigidness and consistency of this draft
The moderating role of emotional regulation on the relationship between school results and personal characteristics of pupils with Attention Deficit/Hyperactivity Disorder
Over time, studies have been conducted that measured the executive functions and emotional regulation of children with ADHD through questionnaires and neurocognitive tasks [26, 27], but there is a need for studies that reflect the unique contribution of both executive functions and emotional regulation to the heterogeneity of ADHD symptoms. From all these studies, the holistic approach of these children is missing, on the one hand, and on the other hand, the objectivity of the survey data to reflect at a behavioral, measurable level, the problems of children with ADHD. The personal and clinical characteristics of children with ADHD are highlighted by evidence that would explain the dysfunctions at the brain level in behavioral, measurable terms. Thus, behavioral solutions can be found for a brain dysfunction.
For example, deficits in executive functioning were associated with predominantly inattentive ADHD and with school difficulty, while emotional regulation difficulty was associated with predominantly hyperactive ADHD at preschool age and with predominantly inattentive ADHD at adolescence [26–29].
Children with ADHD who also have difficulties in emotional regulation are also associated with low social skills, a higher probability that the symptoms of hyperactivity will not remit in adulthood, a lower quality of life than children without ADHD. Deficits in executive functioning can impair mental processes that support self-regulation and can delay learning and development. Difficulty with emotional regulation was also associated with fewer years of schooling, lower likelihood of graduating from high school and college, and higher rates of dropping out of school. In addition, emotional regulation could be a protective factor against stress. Children with ADHD who had a high level of emotional self-regulation coped better with stressful situations at school, exams, or in everyday life, as was the case with the COVID-19 pandemic [30–33].
These contradictory results suggest that there are different associations between cognitive and emotional processes with the heterogeneity of ADHD symptoms throughout a person's development. Indeed, longitudinal studies have shown that the presentation of ADHD in children varies enormously from the preschool to adolescent years [29]. Instead, it remains unclear to what extent individual differences in executive functioning and emotional regulation contribute to the heterogeneity of ADHD symptoms during childhood.
Mohamed et al.[34] mohamed in their study explored whether impairments in core processes (processing speed and distractibility) in individuals with ADHD explain impairments in higher order functions, namely executive functions, memory and complex attention. Hierarchical logistic regression analyzes to assess the contribution of core processes to impairments in higher-order functions revealed: deficits in core processes explained 41–43% of deficits in executive functions, 27–29% in memory, and 56–74% in complex attention. In this case, the role of emotional regulation skills in performance in timed trials was not taken into account.
To complement the information from the above study, Anker and colleagues [35] showed in their study that the association between the result of neuropsychological tests of verbal working memory and processing speed and the association between the severity of attention deficits and emotional regulation deficits in people with ADHD were significant. A shortcoming of this research is: the problem of executive functioning and emotional regulation in ADHD was not put in the light of a holistic approach. The variables were grouped sequentially.
Tenenbaum and colleagues [36] studied in their research the associations between deficits in response inhibition, response execution, and emotion regulationusing a multi-method design. One hundred sixty-six children (aged 5-13 years; 66.3% male; 75 with ADHD) completed two conditions (i.e., neutral and fear) of an emotional go/no-go task. Parasympathetic-based regulation was indexed via respiratory sinus arrhythmia (RSA), and sympathetic-based reactivity was indexed via cardiac pre-ejection period (PEP). Overall, children exhibited more difficulty with response execution (i.e., more omission errors, fewer correct go responses) and less difficulty with response inhibition (i.e., fewer commission errors, more correct no-go responses) during the fear condition than the neutral condition. Children with ADHD displayed more difficulty with response execution during the fear condition compared to typically developing youth. Additionally, children with ADHD displayed parasympathetic-based dysregulation (i.e., RSA increase from baseline) and reduced sympathetic-based reactivity (i.e., PEP lengthening) compared to typically developing youth across task conditions. In sum, children with ADHD demonstrate greater difficulty with response execution during emotionally salient contexts, as well as parasympathetic-based emotion dysregulation. A possible limitation of this research is that it only took into account fear and behavior management in situations where the level of fear is high. The child, on the other hand, in the school environment has a wide range of emotions that he should manage. In addition, behavioral inhibition (postponement of emotional response, identification of emotion and emotional response), planning and organization of behavior in order to achieve school objectives are involved in the learning process. In this study, the difficulty of planning and organizing the emotional and cognitive responses of children with ADHD was not looked at globally.
Morris and colleagues [37] in their research utilized a positive and negative emotion induction and suppression task, as well as indexes of autonomic nervous system reactivity, to examine emotional functioning in youth with ADHD. This study revealed inflexible parasympathetic-based regulation across emotion conditions among youth with ADHD compared to typically developing youth. Future studies should consider the efficacy of adding an emotion regulation skills training component. This study provides the biological basis for the importance of emotional regulation in the functionality of children and young people with ADHD, it is only the starting point for the integration of emotional regulation skills in the context of school performance.
All the mentioned studies approach the emotional regulation and executive functioning of children with ADHD in a restrictive manner. There are studies that show the importance of emotional regulation at the behavioral level by explaining its biological aspects and there are studies that associate emotional regulation only with certain variables such as working memory. It is necessary to take a holistic approach to all the functional difficulties of the child with ADHD in such a way that the child with ADHD can integrate and adapt to the school environment.
- Problems about the reliability and validity of the proposed measures
In order to identify difficulties of emotional regulation (difficulties in recognizing facial expressions, identifying emotions, identifying emotional responses), an image system was developed.
This imaging system was validated in a group of 230 children with neurotypical development before its application to the samples in this study. The fidelity expressed by the Crombach alpha index was 0.87, which allowed its use in this study. The test-retest reliability indicated a value of 0.83 at an interval of 6 months.
The test contains 60 items grouped into 3 scales Recognition of facial expressions, Identification of emotions and Identification of emotional response. Content validity was ensured by the selection of images/items based on studies in the field (the relationship between brain and behavior, the identification and understanding of emotions and the social contexts in which they appear) by two experts in child psychological assessment and child development [34-38]. Content adjustment was based on the same principle as selection. The subsequent improvements of the test were based on the clinical and scientific experience gained by the authors with a series of children. This instrument was revised very carefully 3 times in terms of content and after the pilot test, it was included in this study and the psychometric characteristics were established.
Construct validity was established by checking the correlations between the items of the subscales and subscales that measure similar contents. In our pilot group there were medium correlations (0.4-0.6) between subscales and high correlations between the items of each subscale with the respective subscale (0.7-0.8). The pattern of correlations between each subscale and its items provides clear evidence that the structure of the entire test is solid.
- The role / importance of “emotional regulation” is vague, thus calling for more explanations and supports
The present study aimed to explore the possible moderator role of emotional regu-lation in the relationship between problem-solving ability, visuomotor precision and visuospatial integration, on the one hand, and the school results, on the other hand, in pupils with ADHD. By our knowledge, this is the first study which investigated the role of all these factors in the learning process of children with ADHD, using objective measures.
Based on specialized literature, we can state that difficulties in emotional regulation directly determine difficulties in adjusting the emotional state, which can lead to behaviors inconsistent with the achievement of objectives. Consequently, a possible cause of school problems in ADHD students is difficulty using self-regulatory learning behaviors essential for successful academic performance. Thus, we can justify the moderator role of emotional regulation in the relationship between school results and personal characteristics of students with ADHD [38-41].
Interestingly, inattentive and hyperactive/impulsive symptoms both uniquely predicted emotion regulation, even when controlling for executive functioning, and 18%-30% of working memory’s relation with emotion regulation was conveyed via shared associations with these ADHD symptoms. This pattern suggests that working memory is related to emotion regulation at least in part because underdeveloped working memory contributes to the development and severity of ADHD symptoms, which, in turn, predicts emotion dysregulation[42, 43].
Instead, neuroimaging studies suggest that increased attentionrelated neural activity in emotional contexts reflects allocation of cognitive resources for regulatory control. Such results indicate that children’s difficulties in regulating behaviors in emotional contexts may be the result of competition between welldeveloped emotional processes and poorly developed cognitive control systems[44, 45].
- Key contributions have to be drawn more rigidly
In conclusion, as it appears from previous studies, difficulties in inhibiting or postponing a certain emotional response, difficulties in developing an action plan, difficulties in sequencing actions and difficulties in forming a mental representation of tasks through working memory influence the accuracy and precision of the movements and these are reflected in the school results, especially in handwriting tasks or complex tasks that require the completion of several steps in order[58, 59].
Through the present study, we have illustrated the specific role of emotional regulation in the relationship between the personal characteristics of pupils with ADHD and their school results. In addition, up to this moment, to our knowledge, there is no study that has tested emotional regulation using a practical test in the complex context of the child with ADHD. At the same time, the holistic approach of the child with ADHD in the school context does not appear in the research so far. Unlike other studies, our study took into account all the emotions that the child has in the school environment, and the executive functions were tested with neuropsychological tests. We believe that emotions support the behavior of organizing and planning school activities, and all these are supported by the statistical significance of the moderation model in this study. On the other hand, the biological basis of emotional regulation supports it in its role as moderator of the relationship between personal characteristics and school results of pupils with ADHD.
Theoretical and Practical Significance
Our study brings a holistic perspective on the impact of all problems associated with ADHD on school performance of these children. We demonstrated that difficulties in visuomotor precision, visuospatial integration, and working memory, in association with difficulties in emotional regulation negatively influence the school results of children with ADHD. Accordingly, all these factors must be taken into consideration simultaneously in the management plan of these children.
Limitations and Directions for Future Studies
Some limitations of this study must be noted. The major one is its cross-sectional design. Although the direction of the relationships is theoretically founded and supported by previous studies (e.g., (12)), future research should longitudinally investigate all presumed relationships especially with respect to the moderation hypotheses. Secondly, this study was also based on the school results of the pupils, which can sometimes be established on subjective criteria, depending on the perspective of the teaching staff; the usual variance of the activity product analysis method may partially explain some of the results.It could be an advantage to obtain multiple measures of some constructs (e.g., engagement from the teachers’ perspective).
Therefore, a good emotional adjustment can maximize the relationship between school results and personal characteristics of children with ADHD. The regulation of the emotional state can determine the planning and organization of behavior in achieving school objectives. Thus, emotional regulation can be seen as a protective factor, as a basic resource in the motivation for school performance. In this way, we answer many questions related to the specific role of emotional regulation in the learning process. Teachers, knowing all these things, can work on improving the level of motivation of children for school tasks and on improving the level of self-confidence of children with ADHD, who are often affected by social stigma.
- A minor issue for collection
It is strange to see lots of words embedded with “-“ without any meaningful reason (i.e., Line 36 “re-search”, Line 37 “func-tioning”, Line 39 “regula-tion”)
I corrected the text
Round 2
Reviewer 3 Report
1. The reliability and validity of the proposed measures are still not convinced from the
theoretical perspective, thus calling for further explanation
Although the authors have added on further information about how the image system
is developed (and the corresponding support from the pilot data), less is about the
theoretical support of the proposed construct / measures. It is argued that unless a
construct is well developed and examined, can it be hardly applied meaningfully.
Thus, the authors are strongly recommended to provide further detailed in this regard.
Although the authors skip the descriptions about “the appropriateness of the
proposed approach of cutting points and classification results (i.e., segmenting into
three levels as stated in Line 119-120 shown in the previous version)” in this revision,
the whole moderator test and analytical process of this paper is still based upon such
assumption (owing to the fact that all the analytical outcomes shown in each Table
are the same as what are seen in the previous version). In other words, it is strongly
suggested that the appropriateness of the proposed approach of cutting points and
classification results should be clearly reported.
A minor question related to this issue: in Line 207, “…test these component51of the
executive functions.” Please be sure that all the description is correct.
REVIEWER 1 (minor)
Comments and Suggestions for Authors
Correct in abstract: "with ADHD compared to children without this diagnosis. These results can
be used in the development of intervention programs.
Introduction: covers aspects of ADHD, although no mention is made of the relationship between
learning difficulties and school performance or school results.
Method: Baseline executive functions are assessed, but academic outcomes are not alluded to.
Conclusions: They are scarce, it would be necessary to expand and connect more with what has
been done.
REVIEWER 2 (minor)
Comments and Suggestions for Authors
Thank you for giving us an opportunity to review your research.

Author Response
In order to identify difficulties of emotional regulation (difficulties in recognizing facial expressions, identifying emotions, identifying emotional responses), an image system was developed.
This imaging system was validated in a group of 230 children with neurotypical development before its application to the samples in this study. The fidelity expressed by the Crombach alpha index was 0.87, which allowed its use in this study. The test-retest reliability indicated a value of 0.83 at an interval of 6 months.
The test contains 60 items grouped into 3 scales Recognition of facial expressions, Identification of emotions and Identification of emotional response. Content validity was ensured by the selection of images/items based on studies in the field (the relationship between brain and behavior, the identification and understanding of emotions and the social contexts in which they appear) by two experts in child psychological assessment and child development [34-38]. Content adjustment was based on the same principle as selection. The subsequent improvements of the test were based on the clinical and scientific experience gained by the authors with a series of children. This instrument was revised very carefully 3 times in terms of content and after the pilot test, it was included in this study and the psychometric characteristics were established.
Construct validity was established by checking the correlations between the items of the subscales and subscales that measure similar contents. In our pilot group there were medium correlations (0.4-0.6) between subscales and high correlations between the items of each subscale with the respective subscale (0.7-0.8). The pattern of correlations between each subscale and its items provides clear evidence that the structure of the entire test is solid.
Recognition of facial expressions —The pupil looks at a picture of a facial expression and then identifies the same expression given a choice of four pictures.
Identification of emotions —The pupil identifies an emotion from a choice of four pictures.
Identification of emotional response—The pupil matches a picture of an emotion to a picture that elicits that emotion from a choice of four pictures. Each subtest contains 20 items.
A global score is calculated. A score of 1-16 points is defined as a low score, a score of 17-32 points is an average score, and a score of 33-48 points is a high score. These scores were established based on calculated percentiles. It is important to establish performance levels based on percentiles, because in this way you can see where the child is after a test.
Emotion regulation is a component of the broader concept of self-regulation.Predescu et all. [52] claims that self-regulation refers primarily to control and regulation of one’s emotions and overlaps substantially with inhibitory control, an important component of executive functions.
The test is based on the model of emotional regulation as a process developed by Gross in 2002. This model identifies five emotion regulation strategies that occur during different time points in the emotion experience: situation selection, situation modification, attentional deployment, cognitive change, and response modulation. Gross (1998) further divides these strategies into antecedent-focused and response-focused. Antecedent-focused regulation occurs before the emotion is fully experienced or during the emotion experience, whereas response-focused regulation occurs after the emotion has completely developed. So, with response-focused people have already “responded” to the eliciting event and thus have experienced all the emotion component changes. Within response-focused, people can regulate their emotions by trying to change any of the emotion components. They might change their facial expressions and vocal tone, suppress their thoughts, increase or decrease their physiological arousal, and even change their subjective feelings[53]. Therefore, recognizing and identifying emotions in our own person and in the people around us helps us to regulate our behavior before and after the emotion has appeared.
A similar test was carried out by Wiing in 2008[54] Social Emotional Evaluation which, in addition to the scales we mentioned, also tested the understanding of different conflicting social contexts (Understanding Social Gaffes—The student looks at a picture of a social situation and listens to an accompanying audio clip, then he/she identifies whether everyone in the situation behaved appropriately; Understanding Conflicting Messages—The student looks at a picture and listens to the accompanying audio clip. He/she determines whether the situation contains a conflicting message, and if so, identifies the true meaning of the message. The conflicting messages include humor, sarcasm, and lies.). Test-retest reliability was 0.88.